# GEF-H1 Transduces FcεRI Signaling in Mast Cells to Activate RhoA and Focal Adhesion Formation during Exocytosis

**DOI:** 10.3390/cells12040537

**Published:** 2023-02-07

**Authors:** Yitian Guo, Judeah Negre, Gary Eitzen

**Affiliations:** 1Department of Medicine, University of Alberta, Edmonton, AB T6G 2H7, Canada; 2Department of Cell Biology, University of Alberta, Edmonton, AB T6G 2H7, Canada

**Keywords:** Rho GTPase, guanine nucleotide exchange factor, GEF-H1, ARHGEF2, RhoA, mast cell, degranulation, focal adhesion

## Abstract

When antigen-stimulated, mast cells release preformed inflammatory mediators stored in cytoplasmic granules. This occurs via a robust exocytosis mechanism termed degranulation. Our previous studies revealed that RhoA and Rac1 are activated during mast cell antigen stimulation and are required for mediator release. Here, we show that the RhoGEF, GEF-H1, acts as a signal transducer of antigen stimulation to activate RhoA and promote mast cell spreading via focal adhesion (FA) formation. Cell spreading, granule movement, and exocytosis were all reduced in antigen-stimulated mast cells when GEF-H1 was depleted by RNA interference. GEF-H1-depleted cells also showed a significant reduction in RhoA activation, resulting in reduced stress fiber formation without altering lamellipodia formation. Ectopic expression of a constitutively active RhoA mutant restored normal morphology in GEF-H1-depleted cells. FA formation during antigen stimulation required GEF-H1, suggesting it is a downstream target of the GEF-H1-RhoA signaling axis. GEF-H1 was activated by phosphorylation in conjunction with antigen stimulation. Syk kinase is linked to the FcεRI signaling pathway and the Syk inhibitor, GS-9973, blocked GEF-H1 activation and also suppressed cell spreading, granule movement, and exocytosis. We concluded that during FcεRI receptor stimulation, GEF-H1 transmits signals to RhoA activation and FA formation to facilitate the exocytosis mechanism.

## 1. Introduction

Mast cells are tissue-resident immune cells that play an important role in many cellular processes, including wound healing, inflammation, and immune responses [1]. However, they also contribute to allergic disease via hyper-responsive reactions [2]. Mast cells contain numerous cytoplasmic granules that package pre-formed pro-inflammatory mediators and dysregulation of their release during allergic reactions propagate disease. The most potent activation of mast cells by allergens is mediated through the FcεRI signaling pathway, which leads to the robust release of pro-inflammatory mediators stored in granules. This process of regulated exocytosis is called degranulation [3].

Mast cell degranulation is tightly regulated since this process releases potent pro-inflammatory mediators [3]. Allergen binding to IgE leads to the aggregation of the IgE receptor, FcεRI, on the surface of mast cells. This triggers a downstream signaling cascade via the Lyn-Syk-LAT-PLCγ and the Fyn-Gab2-PI3K signaling pathways [4,5,6,7]. Studies from our lab and others have revealed that Rho GTPases are downstream targets of FcεRI signaling and part of the regulatory mechanism of mast cell degranulation [8,9,10,11]. Rho proteins are monomeric G proteins belonging to the Ras superfamily of GTPases that play diverse roles in many cellular processes, particularly those involved in cytoskeletal dynamics [12]. We have shown that antigen activation of mast cells triggers profound morphological transitions that generate cell protrusions which require Rho GTPase function [10,11]. Generation of this activated cellular state correlates with degranulation and we have suggested that these protrusions are exocytosis zones.

Rho proteins cycle between active GTP-bound and inactive GDP-bound states. Rho guanine nucleotide exchange factors (RhoGEFs) facilitate the loading of GTP, which results in Rho protein activation. RhoGEFs are a diverse family of proteins consisting of 71 Dbl (diffuse B-cell lymphoma) and 11 DOCK (dedicator of cytokinesis) subfamily members in mammals [13,14]. A role for RhoGEFs in exocytosis and membrane trafficking has previously been shown [15,16,17,18,19]. Vav1 and P-Rex1, two RhoGEFs predominantly expressed in hematopoietic cells, were shown to regulate mast cell degranulation [15], GLUT4 protein trafficking in adipocytes [18], and dense granule secretion from platelets [19]. β-PIX, a Rac and Cdc42 RhoGEF, modulates mastoparan-activated mast cell exocytosis and also has been shown to regulate microtubule dynamics in activated mast cells [16,17]. DOCK5, a Rac GEF from the DOCK subfamily, regulates the remodeling of the microtubule network that is essential for mast cell degranulation [20]. GEF-H1, a RhoA GEF, is involved in the processes of membrane trafficking in B cells and epithelial cells [21,22]. These studies depict how RhoGEFs act as signal transduction modulators; however, the exact role of RhoGEFs in exocytosis has not been fully elucidated. 

Here, we investigated the role of the RhoGEF, GEF-H1/ARHGEF2, in granule exocytosis (degranulation) using RBL-2H3 cells, a widely used model of mucosal mast cells [23]. We found that GEF-H1 activation was linked to the FcεRI signaling pathway. Depletion of GEF-H1 significantly disrupted many of the features of activated mast cells, including cell spreading and degranulation. RhoA was found to be the main target of GEF-H1 activation; active RhoA and stress fiber formation were suppressed in GEF-H1-depleted cells. Focal adhesion (FA) formation was found to occur during granule exocytosis and depletion of GEF-H1 also resulted in reduced FA formation after antigen stimulation. Our data shows that the GEF-H1-RhoA signaling axis transduces antigen stimulation signals from FcεRI to facilitate mechanisms contributing to granule exocytosis in mast cells.

## 2. Materials and Methods

### 2.1. Cell Culture and GEF-H1 Knockdown

RBL-2H3 cells were maintained in Eagle’s Minimum Essential Media (MEM) supplemented with 10% (*v*/*v*) heat-inactivated Fetal Bovine Serum (HI-FBS) and 1X antibiotic-antimycotic (1X: 100 units/mL of penicillin, 100 µg/mL of streptomycin, and 0.25 µg/mL of amphotericin B) in a humidified incubator set to 5% CO_2_ and 37 °C. GEF-H1-depleted RBL-2H3 cells were generated by infecting cells with lentivirus (MOI of 10) encoding an shRNA against GEF-H1 mRNA (target region 2115–2135). The virus prepared using the empty vector, FUWG, was used for control cells [24]. GEF-H1 KD and control strains stably expressing shRNA were selected by adding 0.5 µg/mL of puromycin to the media. Strains were used after three passages in the presence of puromycin.

### 2.2. Small Molecule Inhibitors and Antibodies

Small molecule inhibitors used in experiments were the microtubule ligands taxol and nocodazole (Tocris, Bristol, UK), the focal kinase (FAK) inhibitor, PF-573228 (Cayman Chemical, Ann Arbor, MI, USA), and the Syk inhibition GS-9973 (Sigma, St. Louis, MO, USA). All small molecule inhibitors were dissolved in DMSO and used at the concentrations indicated in the figure legends. 0.1% DMSO was used for mock treatment. Cells were pretreated 20 min prior to performing degranulation assays or immunofluorescence microscopy. Commercial antibodies were obtained for CD63 (clone AD1, BioRad, Hercules, CA, USA), GEF-H1 (GenTex, Zeeland, MI, USA), phospho(Ser886)-GEF-H1 (clone E1L6D, Cell Signaling Technology, Danvers, MA, USA), Rac1 (clone 23A8, Sigma, St. Louis, MO, USA), RhoA (clone 26C4, Santa Cruz), β-tubulin (clone EPR16774, Abcam, Cambridge, UK), HA (clone HA-7, Abcam, Cambridge, UK), and vinculin (Proteintech, Rosemont, IL, USA). 

### 2.3. RNA Isolation and qPCR

Total RNA was extracted with Trizol (Invitrogen, Waltham, MA, USA), following the manufacturer’s instruction. Complementary DNA (cDNA) was synthesized from mRNA using oligo dT primers. Briefly, total RNA was extracted from 3 million cells by adding 1 mL of Trizol, then 200 µL of chloroform followed by centrifugation at 12,000× g for 10 min at 4 °C. RNA was precipitated from the top aqueous phase by adding an equal volume of isopropanol, washed in 70% ethanol, and 5 µg was used to synthesize cDNA using 100 units SuperScript™ II Reverse Transcriptase (Invitrogen, Waltham, MA, USA) and 0.5 µg Oligo (dT)12–18 primer (Invitrogen, Waltham, MA, USA) in a 20 µL reaction. To verify the knockdown effects of GEF-H1 shRNA, qPCR was performed using the SensiFAST™ Probe No-ROX kit (Meridian Bioscience, Cincinnati, OH, USA). The qPCR primers for GAPDH were 5′-ACTCCCATTCTTCCACCTTTG and 5′-CCCTGTTGCTGTAGCCATATT, and for GEF-H1 they were 5′-TGTACCAAGGTCAAGCAGAAG and 5′-GCTCTCTGGTGGTTGTCTTAC. For qPCR, a two-step thermocycling reaction was performed based on the Mastercycler^®^ ep realplex Real-time PCR System (Eppendorf, Hamburg, Germany). The 2^−∆∆Ct^ method was used to quantify the mRNA levels with GAPDH as a control [25].

### 2.4. Plasmid Preparation and Transfection of RBL-2H3 Cells

Lifeact-mRuby plasmid was used to label F-actin [26]. P_CMV_-3xHA-RhoA-G14V cloned in pcDNA3.1+ was obtained from the cDNA Resource Center (cDNA.org). A GEF-H1-RNAi resistant mutant construct (GEF-H1-RNAi-Resi) was cloned for re-introduction experiments after the knockdown of the endogenous GEF-H1 mRNA. Full-length GEF-H1 was cloned from RBL-2H3 cell cDNA using the Phusion polymerase (Invitrogen, Waltham, MA, USA) and the forward and reverse primers, respectively, 5′-TCTAAGCTTGTATGTCTCGGATCGAATCCCT and 5′-AGTGGTACCTTAGCTCTCTGAGGCCGTAG. Full-length GEF-H1 was subcloned into the plasmid pmCherry-C1 after Hind3-Kpn1 digestion. This clone of GEF-H1 was used as a template for GEF-H1-RNAi-Resi cloning. The RNAi-resistant primer was designed as follows: forward: CGGAGAGGCCAGAACCTTTAACGGATCCATTGAGCTCTGTAG, reverse: CTACAGAGCTCAATGGATCCGTTAAAGGTTCTGGCCTCTCCG. These primers contained a BamH1 site (underlined) for subsequent verification. A Phusion PCR was performed according to the site-directed mutagenesis strategy previously described [27]. After the transformation of bacteria, clones were selected that incorporated the BamHI site and subsequently verified by Sanger sequencing. Electroporation was used to transfect RBL-2H3 cells with plasmids [28]. A total of 2 million RBL-2H3 cells were mixed with 10 µg purified plasmid in a 400 µL ice-cold electroporation buffer (137 mM NaCl, 2.7 mM KCl, 1 mM MgCl_2_, 10 mM glucose, 20 mM HEPES, pH 7.4). The cell plasmid suspension was then transferred to a 4 mm electroporation cuvette and pulsed by an electric shock at 250 V voltage and 950 µF capacitance (Harvard Apparatus BTX ECM600 Electro Cell Manipulator). Cells were recovered in a complete medium for 24 h and were used for immunofluorescence or live-cell microscopy.

### 2.5. Degranulation Assay

The release of β-hexosaminidase was used for the degranulation assay, as previously described [29]. A total of 100,000 RBL-2H3 cells were plated into wells of a 24-well dish one day before assaying. Cells were sensitized by incubation with 120 ng/mL of anti-DNP IgE (SPE-7, Sigma, St. Louis, MO, USA) for 4 h. Cells were antigen-stimulated by removal of media and the addition of 25 ng/mL of DNP-BSA (ThermoFisher, Waltham, MA, USA) in 200 µL HTB (HEPES-Tyrode’s buffer: 120 mM NaCl, 5 mM MgCl_2_, 1.5 mM CaCl_2_, 1 g/L glucose, 1 g/L BSA, 25 mM HEPES, pH 7.4). β-hexosaminidase levels were determined by an enzyme assay. 100 µL of cell supernatant or an equal fraction of cell lysate (HTB + 0.5% (*v*/*v*) Triton X100) was incubated with 100 µL of 1.2 mM 4-methylumbelliferyl N-acetyl-β-D-glucosaminide (MUG) (Sigma, St. Louis, MO, USA) in a 50 mM citrate buffer, pH 4.5, for 30 min at 37 °C. Reactions were terminated by the addition of 50 µL of 0.2 M glycine, pH 10. Cleavage of MUG by β-hexosaminidase releases the fluorescent product 4-methylumbelliferone, which was detected with a Synergy-4 fluorometer set to 360 nm +/− 20 nm excitation and 450 nm +/− 20 nm emission (BioTek Instruments, Winooski, VT, USA). The level of fluorescence is directly proportional to β-hexosaminidase. Exocytosis was calculated as the percentage of β-hexosaminidase in the supernatant, divided by total β-hexosaminidase as determined from Triton X-100 lysed cells + supernatant.

### 2.6. Detection of Active Rho Proteins and GEF-H1 by Pulldown Assay

The glutathione S-transferase (GST)-tagged protein probes for active GEF-H1 (GST-RhoA-G17A), Rac1 (GST-PAK1-Cdc42/Rac-binding domain), and RhoA (GST-Rhotekin-Rho-binding domain) were prepared by expressing proteins in *E. coli* strain Rosetta™ (DE3) (MilliporeSigma, Burlington, MA, USA) [30,31,32]. RhoA-G17A is a nucleotide-free RhoA mutant that has a high affinity for cognate RhoGEFs; thus, it can be used to detect RhoGEF activation [30]. 0.1 mg of GST-tagged probe, or GST for the control, was immobilized on glutathione agarose, then incubated with cell lysates prepared after various stimulations. Lysates were prepared from 10 million cells in buffer A (20 mM HEPES pH 7.5, 1% (*v*/*v*) Triton X-100, 10 mM MgCl_2_, 150 mM NaCl, 5% (*v*/*v*) glycerol, 2 mM Na_3_VO_4_, 5 mM NaF, 10 µg/mL leupeptin, 10 µg/mL pepstatin, 5 mM *o*-phenanthroline, 0.2 mM PMSF). Proteins were eluted from the resin by adding a 60 µL Laemmli SDS-PAGE buffer and boiling for 5 min. Eluted fraction and a sample of the total lysate were subjected to SDS-PAGE and immunoblot with GEF-H1-, Rac1-, and RhoA-specific antibodies.

### 2.7. Microscopy

Immunofluorescence microscopy was used to examine the intracellular distribution of granules, cytoskeletons, or the localization proteins in RBL-2H3 cells. Cells grown on coverglass were fixed with 4% (*wt*/*v*) paraformaldehyde (PFA) at room temperature (RT) for 30 min, then permeabilized with 0.2% (*v*/*v*) Triton-X100 for 15 min. Cells were blocked with 1% bovine serum albumin (BSA) dissolved in PBS, then incubated with primary antibodies for 2 h at room temperature. Cells were washed 5 times with PBS. Alexa Fluor-conjugated secondary antibodies diluted 1:1000 were used as indicated. Oregon green 488 or Alexa 546 conjugated phalloidin diluted 1:2000 was used to stain F-actin and DAPI (4′, 6-diamidino-2-phenylindol) was used to stain nuclei. Cells were mounted on glass slides with ProLong™ Gold Antifade mounting media (ThermoFisher, Waltham, MA, USA). Images were captured by a Zeiss Observer Z1 microscope (Carl Zeiss, Oberkochen, Germany) with a 63X objective (1.4 NA) and processed using Axiovision 4.8 software.

Live-cell imaging was used to visualize the dynamics of granule trafficking by fluorescence microscopy using Lysotracker Red (ThermoFisher, Waltham, MA, USA) cell morphology transitions in a bright-field [11] and F-actin remodeling F-actin by LifeAct-mRuby [26]. Briefly, previously manipulated RBL-2H3 cells (e.g., Lifeact-mRuby transfected, GEF-H1-depleted or sensitized with anti-DNP-IgE) were grown on round coverslips. Coverslips were placed in an Attofluor chamber (ThermoFisher, Waltham, MA, USA) and growth media was replaced with HTB and placed on a 37 °C-heated microscope stage and objective. Images were captured using a PerkinElmer Ultra-VIEW VoX spinning disk confocal microscope (Waltham, USA) with a 63X objective (1.4 NA) using a 10 s imaging interval. After 1 min of imaging, resting cells were stimulated by the addition of 25 ng/mL of DNP-BSA, and drugs or DMSO were added at the same time. Volocity 6.0 software was used to record and analyze the live-cell videos, which were exported as Window Media files at 10 frames/s.

### 2.8. Analysis of Focal Adhesions

Focal adhesions (FAs) were visualized by immunofluorescence microscopy using vinculin antibodies to label them. We used a method for enrichment of FA staining and quantification that was previously described [33]. Briefly, RBL-2H3 cells were grown on coverslips and then treated with 2.5 mM triethanolamine hypotonic buffer for 3 min at room temperature. Cell bodies were removed by hydrodynamic force using a Waterpik^®^ WP-100 Ultra water flosser for 10 s. The Waterpik was set to 3 and the nozzle was held ~0.5 cm above cells at a 90° angle to flush the cells. The cell body and nuclei were removed by washing and the FA fraction remained bound to the coverslips. Next, the FAs were fixed with 4% (*wt*/*v*) PFA and then labeled with a 1:100 dilution of vinculin antibody. The coverslips were mounted and fluorescent images of identical exposure were captured. FAs were quantified using ImageJ (National Institutes of Health, USA) by measuring the total fluorescence intensity of stained FAs within an individual cell contour [33].

### 2.9. Cell Size Measurement by ImageJ

RBL-2H3 cells undergo spreading and actin remodeling when stimulated [11,23]; thus, the degree of cell spreading can be regarded as an indicator of mast cell activation [34]. The measurement of cell size was performed using ImageJ to analyze the F-actin outlining the cell periphery. Briefly, the selected area of an RGB image was color-thresholded and cells were then outlined using phalloidin-stained areas. Next, the outlined region was automatically analyzed in ImageJ using the “Analyze” feature with the output values of area, mean, and integrated density.

### 2.10. Statistical Analysis

Quantified data are shown as mean ± s.e.m. (standard error of the mean). Student’s *t*-test was used to identify statistically significant differences between responses from data with two independent variables and one-way ANOVA for parametric data with three or more independent variables. A post hoc analysis by Tukey’s HSD (honestly significant difference) test was used to identify pairwise significant differences. Differences in non-parametric data (i.e., cell size) were analyzed by Kruskal–Wallis with a post hoc analysis by Dunn’s tests. Statistical analyses were performed using Microsoft Excel Xrealstats Add-in (Real Statistics). *p*-values < 0.05 were considered statistically significant.

## 3. Results

### 3.1. Establishment of a Role for GEF-H1 (ARHGEF2) in Mast Cell Degranulation

Previous studies have shown that Rho proteins, such as Rac1 and RhoA, are involved in mast cell granule exocytosis [9,10,11,35,36]. Rho proteins are activated by Rho guanine-nucleotide exchange factors (RhoGEFs), a class of proteins that transduce receptor signaling to downstream Rho protein activation. We hypothesized that GEF-H1 may be a putative RhoGEF involved in regulating mast cell granule exocytosis because it has been shown to activate both Rac1 and RhoA [37,38,39], and it is associated with the exocytosis complex called the exocyst [21,22]. We generated an RBL-2H3 cell line depleted of GEF-H1 (GEF-H1 KD) using lentivirus-mediated shRNA knockdown. qPCR and immunoblot analysis were used to verify depletion of GEF-H1 mRNA and protein, respectively. GEF-H1 mRNA levels were reduced by 81.3% ± 7.3% compared to control cells (Figure 1A) and immunoblot of lysates showed protein levels were similarly reduced (Figure 1B). The effect of GEF-H1 KD on mast cell granule exocytosis was examined by degranulation assay. Background levels of exocytosis were similar in all cell lines (Figure 1C, time 0). However, granule exocytosis was significantly reduced in GEF-H1 depleted cells when antigen-stimulated for 15 min and 30 min (Figure 1C). These results suggest GEF-H1 may have an important regulatory role in the mast cell granule exocytosis mechanism.

### 3.2. GEF-H1 Knockdown Results in Reduced Cell Activation and Granule Trafficking

The activation of RBL-2H3 cells by antigen stimulation leads to cell spreading [11,40], which indicates cytoskeletal remodeling is actively occurring. Cell activation also results in the redistribution of secretory granules toward the cell periphery [41,42]. We used immunofluorescence and live-cell microscopy to determine the cellular effects of GEF-H1 depletion on granule trafficking and cell morphology. Control RBL-2H3 cells underwent normal spreading and granules were widely dispersed in the cytoplasm after antigen stimulation (Figure 2A, control). In GEF-H1-depleted cells, antigen stimulation resulted in significantly less cell spreading; however, granules seemed to be well dispersed in the cytoplasm (Figure 2A, GEF-H1 KD). Closer examination revealed that granules were enriched adjacent to the plasma membrane in higher abundance in GEF-H1-depleted cells (Figure 2A, right panels), which is consistent with an observed reduction in degranulation (see Figure 1C). Quantification of cell size showed no difference prior to stimulation; however, GEF-H1-depleted cells showed a significant difference in size after 30 min of antigen stimulation compared to RBL-2H3 and control cells (Figure 2B). The average area of control cells increased 4.2-fold after stimulation, while GEF-H1-depleted cells increased only 1.9-fold. This supports the notion that GEF-H1 may regulate cell morphology, creating a flattened state with more surface area which facilitates granule exocytosis.

To confirm that the loss of cell spreading and granule distribution can be attributed to the depletion of GEF-H1, we examined whether the reintroduction of GEF-H1 into knockdown cells would rescue these defects. For this, we made an RNAi-resistant construct, GEF-H1-Resi, that was tagged with mCherry. The mCherry-C1 empty vector was used as a control. In control cells, expression of GEF-H1-Resi or mCherry-C1 did not alter cell morphology (Figure 3A). In GEF-H1-depleted cells, the expression of GEF-H1-Resi restored cell spreading, while the expression of mCherry-C1 did not show any rescue effect (Figure 3B). Analysis of cell size confirmed that, upon antigen stimulation, depletion of GEF-H1 prevented cell spreading, which was restored to normal levels by expression of GEF-H1-Resi (Figure 3C). This rescue of defects in GEF-H1-depleted cells confirms that GEF-H1 plays a role in regulating cell morphology transitions that occur in stimulated RBL-2H3 cells.

We used live-cell imaging to examine the role of GEF-H1 in the dynamics of cell morphology transitions and granule trafficking during antigen stimulation. Cell morphology was imaged by bright-field microscopy and granules were labeled with LysoTracker Red and imaged by spinning-disk confocal microscopy. Videos show control cells first formed dorsal membrane ruffles then large lamellipodia, causing cells to spread and flattened with granules projecting into the flattened areas (Appendix A). GEF-H1 KD cells also formed dorsal ruffles soon after stimulation; however, cells did not form large lamellipodia and did not spread. Granules were found to accumulate at the plasma membrane (Appendix A). Still, images extracted from videos show granules accumulating at the periphery of GEF-H1-depleted cells, while few granules accumulated at the plasma membrane in control cells (Figure 4A, arrows). Granule tracking analysis revealed that GEF-H1 may affect the velocity of granules, which normally increase after stimulation (Figure 4B). Depletion of GEF-H1 resulted in a 30% reduction in granule velocity, 0.578 +/− 0.0518 µm/s compared to 0.827 +/− 0.0829 µm/s in control cells. The reduced motility of secretory granules and their peripheral accumulation in GEF-H1-depleted cells is consistent with a reduction in granule exocytosis.

### 3.3. RhoA, but Not Rac1, Is a Downstream Target of GEF-H1

GEF-H1 has previously been reported to be a RhoGEF for RhoA and Rac1 [37,38,39,43]. To determine the downstream activation target(s) of GEF-H1 in mast cells, we used a pulldown assay with GST-tagged Rhotekin and PAK1 Rho-binding domain probes that bind to active RhoA-GTP and Rac1-GTP, respectively [31,32]. In control cells, antigen stimulation increased the levels of active Rac1-GTP and RhoA-GTP (Figure 5A,B, respectively). However, in GEF-H1-depleted cells, antigen stimulation resulted in no increase in active RhoA-GTP levels, while active Rac1-GTP levels increased similar to that observed in control cells (Figure 5A,B, respectively). These results suggest RhoA activation is the downstream target of GEF-H1, since the knockdown of GEF-H1 prevented the activation of RhoA, but not Rac1, after antigen stimulation.

RhoA regulates the formation of stress fibers in various cells [44,45]; therefore, stress fiber formation can be considered a physiological indicator of RhoA activity. Stress fiber formation was examined in the antigen-stimulated RBL-2H3 control and GEF-H1-depleted cells. There were few stress fibers observed in cells prior to stimulation (Figure 5C, panels a and d). However, when antigen-stimulated, control cells formed prominent stress fibers across the cell (Figure 5C, panels b and c, red arrows), while GEF-H1-depleted cells lacked similar stress fiber formations (Figure 5C, panels e and f). This result supports the conclusion that RhoA activation is controlled by GEF-H1 during antigen stimulation. Stress fiber formation could facilitate the projection of the leading edge of cells for cell spreading, which was significantly reduced by GEF-H1 depletion (see Figure 3).

Rac1 activation triggers lamellipodia formation and cell ruffling [44,45]. To rule out Rac1 as a possible downstream target of GEF-H1, live-cell imaging was used to visualize the dynamic formation of lamellipodia that occurs during RBL-2H3 stimulation [11]. Live-cell imaging via differential interference contrast (DIC) microscopy showed that membrane ruffling occurred in both the control cells (Appendix A) and GEF-H1-depleted cells (Appendix A). Furthermore, actin remodeling was directly imaged in live cells using the F-actin probe, Lifeact-mRuby. This showed that antigen stimulation triggered the formation of lamellipodia at the leading edge of control cells (Appendix A) and similarly in GEF-H1-depleted cells (Appendix A). This suggests that Rac1 activation is maintained in the absence of GEF-H1. These observations are in agreement with results showing Rac1 activation is maintained (see Figure 5A). Taken together, these results suggest that Rac1 was not a downstream Rho protein regulated by GEF-H1 in RBL-2H3 cells during antigen stimulation.

### 3.4. Expression of Constitutively Active RhoA Bypasses GEF-H1

To further examine whether the effects of GEF-H1 depletion were due to a lack of RhoA activation, we transfected cells with a constitutively active RhoA mutant, RhoA-G14V, to determine if defects could be rescued. Control and GEF-H1-depleted cells were transfected with a 3×HA-tagged RhoA-G14V expressed from a CMV promoter, or empty vector for the control. Cells were either left unstimulated or stimulated for 30 min, and then fixed and stained with anti-HA to mark transfected cells, anti-CD63 to mark granules, and Alexa Fluor 405-phalloidin to show cell morphology. Control cells transfected with either vector or RhoA-G14V resulted in granules that were well dispersed and cells that spread after antigen stimulation (Figure 6A, upper two rows). In GEF-H1-depleted cells, transfection with RhoA-G14V restored normal granule distribution and cell spreading after antigen stimulation, while transfection with an empty vector did not (Figure 6A, bottom two rows).

Quantification of cell area showed that control cells were significantly larger than GEF-H1-depleted cells when transfected with an empty vector; however, the ability to spread and increase in size was restored to normal levels by transfection of GEF-H1-depleted cells with RhoA-G14V (Figure 6B). These results show that defects due to GEF-H1 depletion can be bypassed by expressing constitutively active RhoA and supports the conclusion that RhoA is the downstream target of GEF-H1. Taken together, these data suggest that during antigen stimulation of RBL-2H3 cells, RhoA-GEF-H1 signaling is required for morphological transitions to generate an activated state.

### 3.5. The GEF-H1-RhoA Signaling Axis Regulates Focal Adhesion (FA) Formation

Previous studies have shown that RhoA is a key regulator of focal adhesion (FA) formation [44]. In addition, it was shown that focal adhesion kinase (FAK), a key regulator of FA formation, was activated in antigen-stimulated mast cells [46]. Therefore, we next examined whether FA formation was a downstream target of the GEF-H1-RhoA signaling axis in RBL-2H3 cells. 

The effect of the FAK inhibitor, PF-573228, on granule exocytosis was examined by a degranulation assay. PF-573228 had no effect on basal levels of degranulation but showed significant inhibition of degranulation after antigen stimulation (Figure 7A). PF-573228 also prevented cell spreading and granule dispersion (Figure 7B). This suggests that inhibition of FA formation by PF-573228 disrupted the cell activation mechanism that leads to granule trafficking during antigen stimulation. The number of FAs formed was examined in unstimulated and stimulated cells by shearing away cell bodies and staining the remaining adherent FAs with vinculin antibody (Figure 7C). Antigen stimulation resulted in an increase in the intensity of FA staining compared to unstimulated cells, while pretreatment with the FAK inhibitor, PF-573228, reduced FA staining (Figure 7D). These results are consistent with the requirement of FAs to support granule exocytosis in antigen-stimulated RBL-2H3 cells. FA formation was also analyzed in GEF-H1-depleted cells. Control cells showed a robust increase in FA staining after antigen stimulation, while GEF-H1-depleted cells did not show a comparably robust increase in FAs (Figure 7E,F). These results show that the depletion of GEF-H1 disrupts the formation of FAs, which suggests that the GEF-H1-RhoA signaling axis may facilitate the generation of an activated mast cell state through FA formation.

### 3.6. GEF-H1 Is Activated in Antigen-Stimulated Mast Cells via the FcεRI Signaling Pathway

We hypothesize that GEF-H1 transduces signals from the cell surface receptor, FcεRI, to downstream Rho proteins. Therefore, we next examined whether GEF-H1 activation is linked to the FcεRI signaling pathway. We performed assays for GEF-H1 activation using GST-RhoA-G17A for affinity precipitation [30]. RhoA-G17A is a nucleotide-free mutant of RhoA that has a high binding affinity for RhoA-specific GEFs. RBL-2H3 cell lysates were incubated with GST-Rho-G17A or GST only bound to glutathione resin and GEF-H1 was found to bind selectively to the GST-RhoA-G17A probe and not GST (Figure 8A). Levels of GEF-H1 binding increased during a time course of antigen stimulation, showing that GEF-H1 activation may be linked to FcεRI signaling (Figure 8B).

GEF-H1 activation has been shown to occur by two distinct mechanisms: microtubule dynamics and phosphorylation. We and others have shown a prominent role in microtubule dynamics in regulating mast cell granule trafficking and exocytosis [42,47,48,49]. GEF-H1 was previously reported to be a microtubule-bound RhoGEF [38,50] that may link microtubule remodeling to the activation of Rho proteins [37,38,39]. Active GEF-H1 was shown to be regulated by release from microtubules [38,50]. When we preincubated cells with the microtubule-stabilizing drug, taxol, there was no change in active GEF-H1 levels, while the microtubule-destabilizing drug, nocodazole, induced a small increase in active GEF-H1 levels (Figure 8C). However, immunoprecipitation of GEF-H1 from RBL-2H3 cells showed no association with tubulin (Figure 8D). These results suggest that while GEF-H1 activation is linked to FcεRI signaling, it might not rely on microtubule dynamics in mast cells and instead may be regulated by phosphorylation.

Syk is an FcεRI proximal kinase that is essential for mast cell degranulation [51]. The Syk-specific inhibitor, GS-9973, potently inhibited antigen-stimulated degranulation with an IC_50_ of ~1 nM (Figure 9A). The Syk inhibitor also showed a dose-dependent inhibition of the morphology transitions associated with RBL-2H3 cell activation (Figure 9B). The Syk-dependent regulation of GEF-H1 activation was demonstrated by the GEF activation assay. Levels of active GEF-H1 increased after antigen stimulation but were significantly reduced when cells were preincubated with 10 µM Syk inhibitor (Figure 9C). These results indicated that activation of GEF-H1 in antigen-stimulated RBL-2H3 cells was Syk-dependent and thus likely regulated by phosphorylation. While GEF-H1 is known to be activated by phosphorylation [43,50,52,53], whether it is a direct substrate of Syk requires further investigation.

## 4. Discussion

Mast cells release potent pro-inflammatory mediators by a highly regulated mechanism of granule exocytosis called degranulation. Aggregation of the IgE surface receptor, FcεRI, results in robust degranulation. Here, we show that the RhoGEF, GEF-H1, is a downstream target of FcεRI signaling involved in regulating processes that facilitate mast cell granule exocytosis. GEF-H1 (also known as ARHGEF2) is a multi-domain protein with a tandem DH-PH (Dbl homology–Pleckstrin homology) domain necessary for Rho protein GTP exchange, an *n*-terminal C1 domain, which suggests it can be regulated by diacylglycerol, and two coiled domains involved in protein interactions. GEF-H1 has been shown to activate both Rac1 and RhoA [37,38,39,54]. However, we found that RhoA is the primary target of GEF-H1 during antigen stimulation in mast cells (Figure 5). RhoA activation was found to be deficient in GEF-H1-depleted cells, as was the formation of stress fibers which require RhoA [44]. Rac1 activation was unaffected and downstream functions of Rac, such as the formation of lamellipodia, still occurred in GEF-H1-depleted cells (Appendix A). Furthermore, the expression of constitutively active RhoA rescued stress fiber formation and exocytosis in GEF-H1-depleted mast cells.

Rho GTPases are signaling molecules well-known to regulate actin cytoskeletal remodeling in response to extracellular stimuli [44,45]. RhoGEFs, the upstream activators of Rho GTPases, are thus likely to be pivotal signal transducers of external stimuli. Several RhoGEFs have been shown to function in signaling pathways leading to exocytosis in mast cells [15,16,17] and various other secretory cell types [18,19,55,56,57]. RhoGEFs couple exocytosis with these morphological transitions since their function has been associated with cytoskeleton remodeling that occurs in conjunction with stimulated secretion [58,59]. GEF-H1, in particular, has been shown to be associated with the plasma membrane exocytosis machinery called the exocyst to control exocytosis [21,22]. While the exocyst is likely a universal component of the exocytosis machinery [60,61], it has not yet been demonstrated to be involved in mast cell degranulation. While our results show a GEF-H1 dependence for mast cell degranulation, we could not detect any interactions with the exocyst complex. Hence, the role of Rho GTPase in exocytosis may be specific to cell morphology transitions needed to facilitate granule docking. Indeed, our previous results showed sequential activation of Rac1 first and subsequently RhoA [11]. It is possible that Rac1-stimulated lamellipodia lead to cell spreading, while RhoA-stimulated stress fibers maintain the activated state. Stress fiber formation could also facilitate cell retraction and initiate a recovery phase back to the resting state.

The role of GEF-H1 in mast cell degranulation may also be due to the stimulation of focal adhesions (FAs). Upon stimulation, RBL-2H3 cells flatten and numerous FAs formed in the spreading area (Appendix A, Figure 7). We show that FAs are a crucial part of the mast cell exocytosis mechanism as the FA kinase inhibitor, PF-573228, also inhibited mast cell degranulation. Depletion of GEF-H1 led to a reduction in FA formation that occurs after antigen stimulation, suggesting that FA formation may be one of the functions of the GEF-H1-RhoA signaling axis. This is consistent with the GEF-H1 activation of RhoA, as FA formation is driven by RhoA signaling [44,62]. Indeed, a RhoA-GEF-H1 signaling network has been shown to drive localized exocytosis at FA sites [63,64]. The resolution of our images was not sufficient to conclude that exocytosis occurred at FAs and thus a specific role for GEF-H1 in this event remains to be determined.

The activation of GEF-H1 relied on Syk kinase which is part of the kinase cascade activated by the aggregation of IgE-FcεRI complexes. Mast cell activation and degranulation can be effectively blocked by the Syk inhibitor GS-9973; this inhibitor was also found to block GEF-H1 activation (Figure 9). While these data link GEF-H1 activation to the FcεRI signaling pathway, they do not show that GEF-H1 is a direct substrate of Syk. Previous studies have shown that GEF-H1 can be activated by either tyrosine or serine/threonine phosphorylation [43,50,53]. GEF-H1 contains an autoinhibitory domain (AID) with a central tyrosine (Tyr198) surrounded by negatively charged and lipophilic residues, which was proposed to interact with the DH (Dbl homology) domain to block its catalytic activity [53]. GEF-H1 can be activated by Src phosphorylation at Tyr198, leading to the unblocking of the DH domain [53]. This is similar to the activation manner of another RhoGEF; Vav1 can be phosphorylated at Tyr174 to dissociate the DH domain from the AID [65]. GEF-H1 can also be inactivated by phosphorylation. It was shown that the knock-out of the serine/threonine kinase, Pak2, which is highly abundant in mast cells, leads to increased mast cell degranulation [66]. Pak2 phosphorylates GEF-H1 at Ser-885, which induces 14-3-3 binding and its inactivation [66]. This result supports our conclusions that RhoA-GEF-H1 signaling plays an important role in the mast cell granule exocytosis mechanism.

GEF-H1 was previously found to be microtubule-bound GEF. The binding of GEF-H1 to microtubules restricted its GEF activity in various cells, as reviewed in [67]. Nocodazole treatment, which dissociates microtubules, led to the release and activation of GEF-H1 [68,69]. Therefore, there seem to be two modes of GEF-H1 regulation: protein phosphorylation and microtubule-dependent regulation. In RBL-2H3 cells, we found that the localization of GEF-H1 was not markedly altered by microtubule-targeted drugs (data not shown). Treating with nocodazole did result in a slight increase in activated GEF-H1 (Figure 8C). However, studies in primary mast cells and Jukat T cells suggest microtubule binding was intrinsic to phospho-regulation, such that phosphorylation affected microtubule interaction [66,70]. Our studies relied on the use of RBL-2H3 cells and this model system may have some limitations for the detection of microtubule regulation. Further studies in other cells or animal systems are needed to validate the impacts of microtubule dynamics and protein phosphorylation in the regulation of GEF-H1.

## Figures and Tables

**Figure 1 cells-12-00537-f001:**
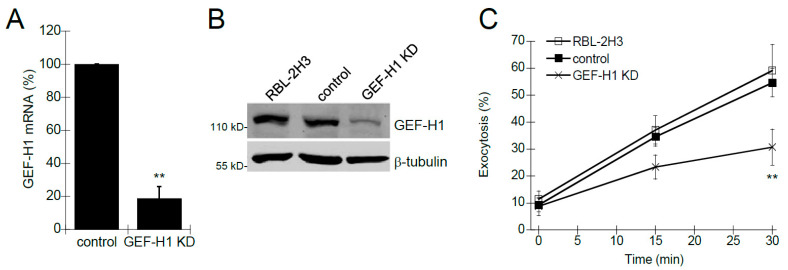
The Effect of GEF-H1 knockdown (KD) on mast cell degranulation. GEF-H1 expression was reduced in RBL-2H3 cells using RNA interference and the effect on degranulation was assayed. (**A**) qPCR was used to quantify levels of GEF-H1 mRNA in control cells infected with the shRNA vector or GEF-H1 shRNA (GEF-H1 KD). Calculated 2^−ΔΔCt^ values were normalized relative to the control. A 81.3% reduction in GEF-H1 mRNA was observed in the GEF-H1 KD strain (** *p* = 0.0090 comparing the reduction in GEF-H1 expression in GEF-H1 KD to the control by one-tailed Student’s *t*-test; *n* = 3). (**B**) Immunoblot was used to confirm the reduction in GEF-H1 protein in the GEF-H1 KD strain. (**C**) Degranulation assays were used to determine the effect of GEF-H1 depletion on mast cell granule exocytosis and were statistically analyzed by one-way ANOVA (*p* < 0.001; *n* = three independent blots). Post hoc Tukey’s tests revealed GEF-H1 knockdown significantly reduced exocytosis in pairwise comparisons between RBL-2H3 and control strains after 30 min of stimulation (** *p* < 0.001 and 0.00314 for RBL-2H3 and control strains, respectively).

**Figure 2 cells-12-00537-f002:**
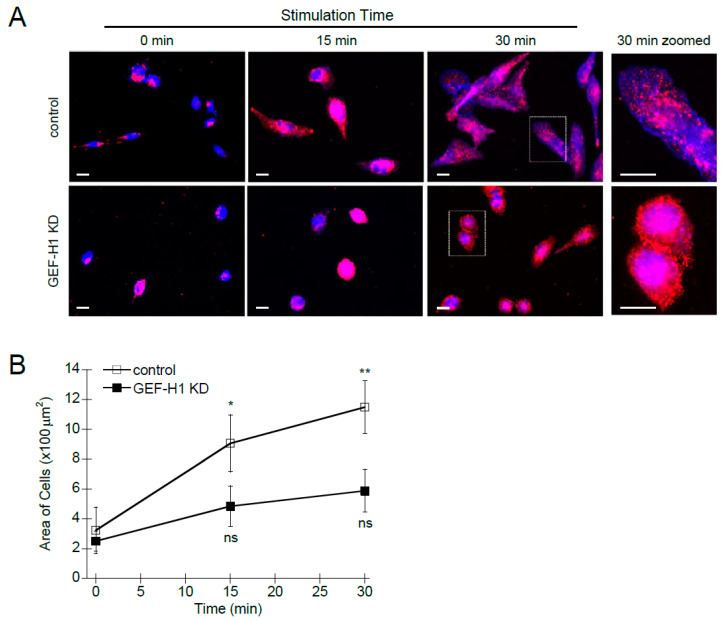
Depletion of GEF-H1 causes cell morphology defects. Control and GEF-H1 knockdown (GEF-H1 KD) RBL-2H3 cells were unstimulated (0 min) or antigen-stimulated for 15 min and 30 min. (**A**) Fixed cells were labeled with an anti-CD63 antibody (red) which labels granules [42], and Alexa Fluor 405-phalloidin which labels F-actin (blue). Scale bar = 20 μm. (**B**) Quantification of cell size by ImageJ analysis. Knockdown of GEF-H1 significantly reduced cell spreading after 15 and 30 min of antigen stimulation. For statistical analysis, the size of 5 to 8 cells per condition (replicate measurements) was averaged from three independent experiments. Differences in cell size between stimulation time points were analyzed by Kruskal–Wallis tests (*p* = 0.00858). Dunn’s post hoc tests were used to identify significant differences between the size of unstimulated and stimulated cells (* *p* = 0.0266 and ** *p* = 0.00591 for control cells stimulated 15 and 30 min, respectively; not significant (ns) for GEF-H1 KD cells).

**Figure 3 cells-12-00537-f003:**
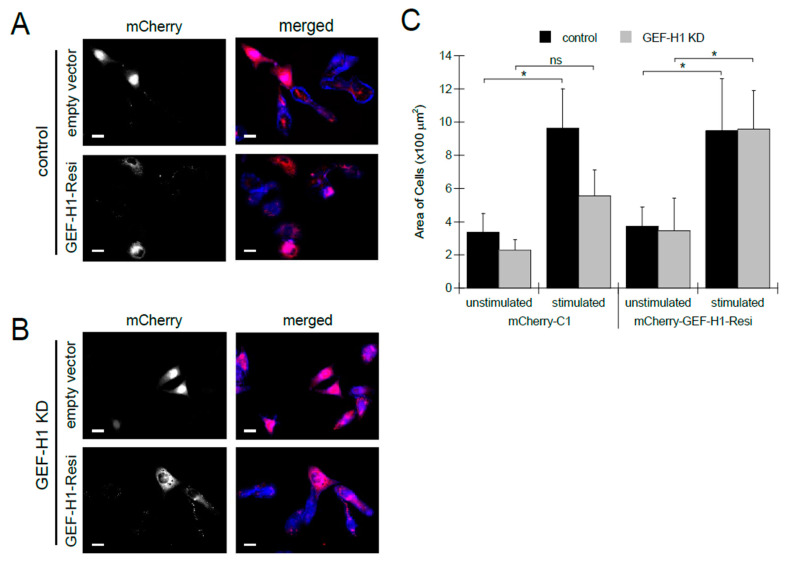
Expression of RNAi-resistant GEF-H1 restores cell spreading in antigen-stimulated GEF-H1-depleted RBL-2H3 cells. The GEF-H1 RNAi resistant construct (mCherry-GEF-H1-Resi) or empty vector (mCherry-C1) was transfected into the control (**A**) and GEF-H1-depleted (GEF-H1 KD) (**B**) cells. (**A**,**B**) Images of RBL-2H3 cells after 30 min of antigen stimulation. Granules are labeled with anti-CD63 antibody (red) and F-actin is labeled with Alexa Fluor 405-phalloidin (blue). Transfected cells (grayscale) are indicated by mCherry expression. Scale bar = 20 μm. (**C**) Quantification of cell size by ImageJ analysis. When stimulated, knockdown of GEF-H1 prevented cell spreading, while the expression of GEF-H1-Resi in GEF-H1-depleted cells restored cell spreading. For statistical analysis, the size of 5 to 8 cells per condition (replicate measurements) was averaged from three independent experiments. Differences in cell size between samples were analyzed by Kruskal–Wallis tests (*p* = 0.00578). Dunn’s post hoc tests were used to identify significant differences between the size of unstimulated and stimulated cells (* *p* = 0.0282 for control and not significant (ns) for GEF-H1 KD cells transfected with mCherry-C1; * *p* = 0.0377 for control and * *p* = 0.0433 for GEF-H1 KD cells transfected with mCherry-GEF-H1-Resi).

**Figure 4 cells-12-00537-f004:**
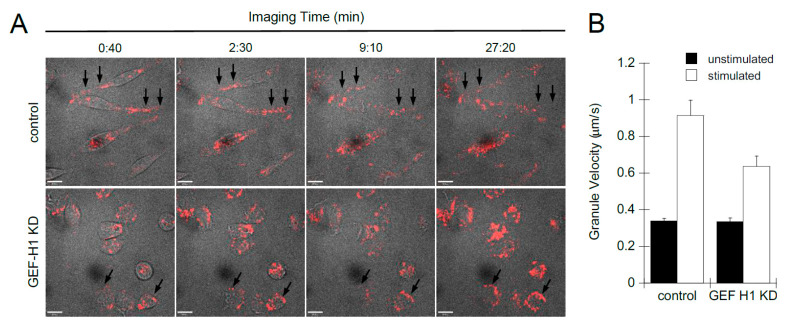
Depletion of GEF-H1 affected cell spreading and granule mobility. Cell morphology transitions and granule mobility were examined by live-cell imaging of RBL-2H3 cells during antigen stimulation. Granules were labeled with LysoTracker Red and cells were stimulated after 1 min of imaging. (**A**) Representative still images of time points from control cells (Appendix A) and from GEF-H1-depleted cells (Appendix A). Compared to control cells, GEF-H1 KD reduced granule motility and caused their retention adjacent to the plasma membrane (arrows). Scale bar = 10 µm. (**B**) Depletion of GEF-H1 resulted in reduced granule velocity. Granule velocity was measured using Volocity v6.3 software (*n* ≥ 37 granules tracked from two independent videos for each strain). Appendix A showing live-cell imaging of LysoTracker Red are available at the following direct object identifiers: Appendix A: LysoTracker Red live-cell imaging of stimulated RBL-2H3 cells; Appendix A: LysoTracker Red live-cell imaging of stimulated RBL-2H3 cells, GEF-H1 knockdown.

**Figure 5 cells-12-00537-f005:**
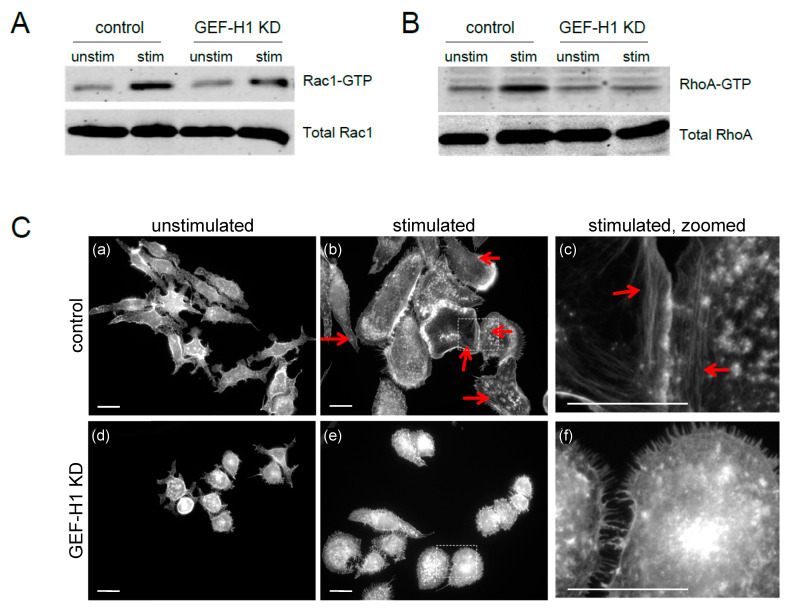
RhoA, but not Rac1, is a downstream effector of GEF-H1 in antigen-stimulated RBL-2H3 cells. Detection of activated Rac1 (**A**) and RhoA (**B**) was performed by pulldown assays using Rhotekin or PAK1 probes that bind RhoA-GTP and Rac1-GTP, respectively. Active levels of both RhoA and Rac1 increased in control cells after 20 min of antigen stimulation (control, stim). Depletion of GEF-H1 did not affect Rac1-GTP levels, but resulted in reduced RhoA-GTP levels (GEF-H1 KD, stim). (**C**) Depletion of GEF-H1 affected the formation of stress fibers. Alexa Fluor 405-phalloidin was used to label F-actin structures in unstimulated cells (left panels) and after 20 min of antigen stimulation (middle panels). Control cells showed the formation of stress fibers that project to the plasma membrane (panels b and c, red arrow). GEF-H1-depleted cells lacked stress fiber formation and showed reduced cell spreading (panels e and f). Scale bar = 20 μm.

**Figure 6 cells-12-00537-f006:**
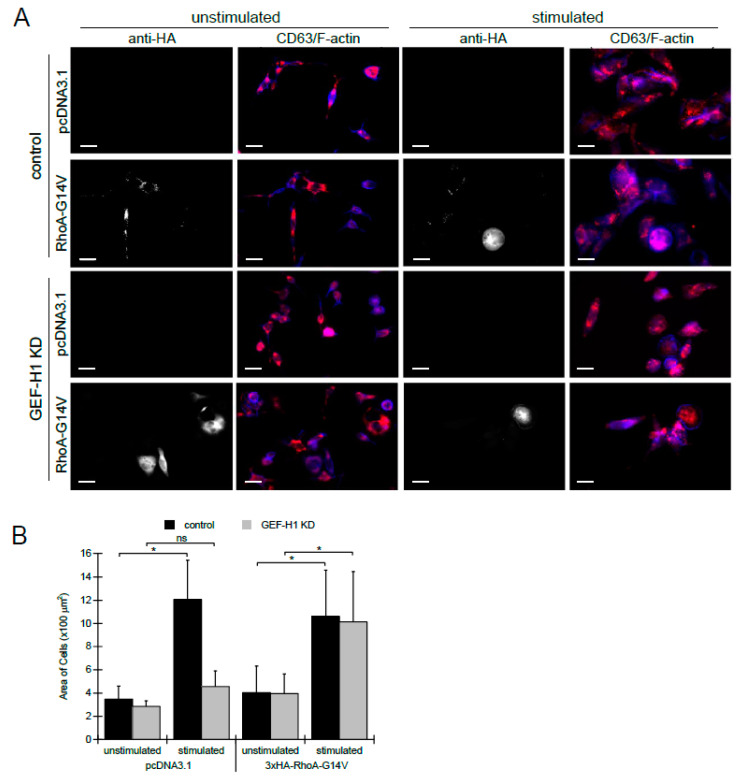
Expression of constitutively active RhoA-G14V in GEF-H1-depleted cells rescues defects in granule trafficking and cell spreading. 3xHA-tagged RhoA-G14V, a constitutively active mutant of RhoA, was transfected into control and GEF-H1-depleted (GEF-H1 KD) RBL-2H3 cells. pcDNA 3.1 was used as an empty vector control. Granules were labeled with anti-CD63 antibody (red) and F-actin was labeled with Alexa Fluor 405-phalloidin (blue). Anti-HA labeling (grayscale) indicated transfected cells. (**A**) Images show that the expression of 3xHA-RhoA-G14V rescues defects of both granule distribution and cell spreading after antigen stimulation (30 min) of GEF-H1-depleted cells. Scale bar = 20 μm. (**B**) Quantification of cell size by ImageJ analysis. When stimulated, the knockdown of GEF-H1 prevented cell spreading, while the expression of RhoA-G14V in GEF-H1-depleted cells restored cell spreading. For statistical analysis, the size of 4 to 6 cells per condition (replicate measurements) was averaged from three independent experiments. Differences in cell size between samples were analyzed by Kruskal–Wallis tests (*p* = 0.00794). Dunn’s post hoc tests were used to identify significant differences between the size of unstimulated and stimulated cells (* *p* = 0.0243 for control and not significant (ns) for GEF-H1 KD cells transfected pcDNA3.1; * *p* = 0.0327 for control and * *p* = 0.0433 for GEF-H1 KD cells transfected with 3xHA-RhoA-G14V).

**Figure 7 cells-12-00537-f007:**
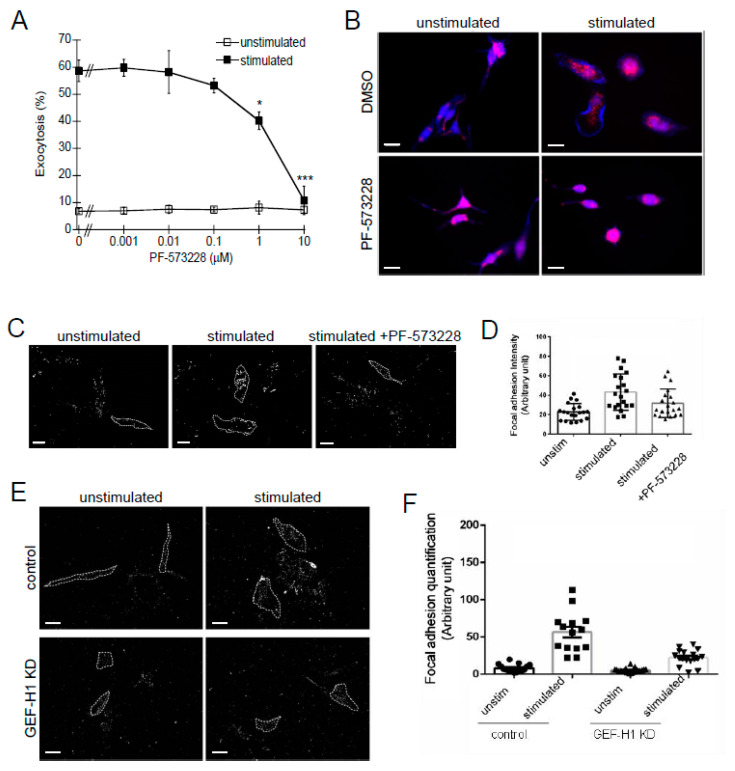
Involvement of focal adhesion (FA) formation in mast cell granule exocytosis. The focal adhesion kinase inhibitor, PF-573228, inhibits granule exocytosis. RBL-2H3 cells were preincubated with 1 nM to 10 μM PF-573228 for 30 min followed by 30 min of antigen stimulation. (**A**) Degranulation assays showed a significant reduction in granule exocytosis when RBL-2H3 cells were pretreated with 1 or 10 µM PF-573228 (* *p* = 0.0488 and *** *p* < 0.001 at 1 and 10 µM PF-573228, respectively, vs. DMSO control by one-way ANOVA and post hoc Tukey’s test; *n* = three independent assays). (**B**) Fluorescence microscopy of RBL-2H3 cells showing granule distribution by LysoTracker Red staining and F-actin by Alexa Fluor 405-phalloidin staining (blue). Inhibition of FA formation with 10 µM PF-573228 prevented granule dispersion and cell spreading. Scale bar = 20 μm. (**C**) FAs were imaged after hypotonic shock by anti-vinculin staining in unstimulated cells or antigen-stimulated cells +/− 10 µM PF-573228. Scale bar = 20 μm. (**D**) Quantification of FA formation after antigen stimulation (*n* ≥ 20 cells from two independent experiments). (**E**) FA formation is reduced in GEF-H1-depleted (GEF-H1 KD) RBL-2H3 cells. Scale bar = 20 μm. (**F**) Quantification of FA formation in antigen-stimulated control and GEF-H1-depleted cells (*n* ≥ 20 cells from two independent experiments).

**Figure 8 cells-12-00537-f008:**
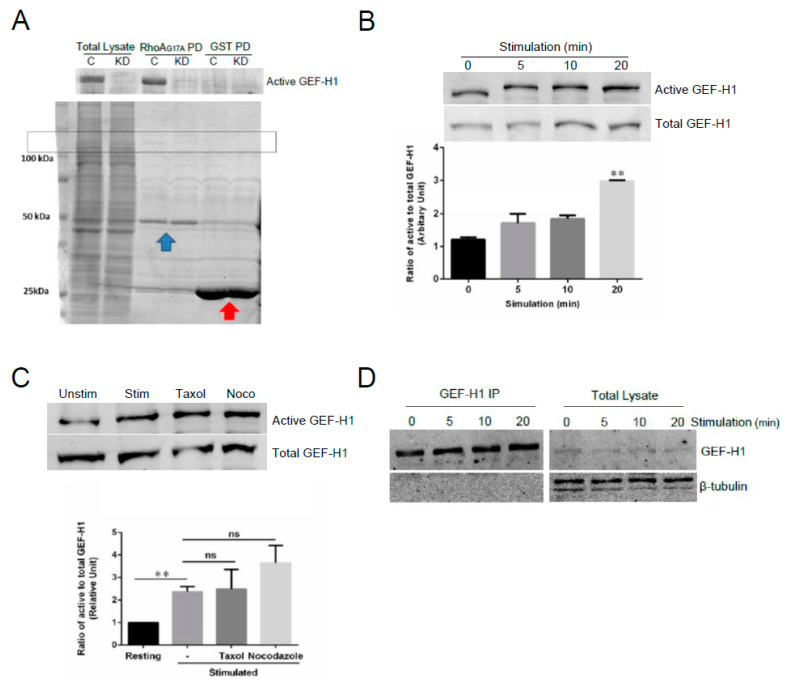
GEF-H1 is activated in RBL-2H3 cells during antigen stimulation. GEF-H1 activation was determined by a pulldown assay using immobilized GST-RhoA-G17A, a nucleotide-free mutant of RhoA that has a high affinity for RhoA-associated active GEFs. (**A**) Detection of active GEF-H1 in RBL-2H3 cell lysates prepared from control (**C**) and GEF-H1-depleted (KD) cells. The total lysate is 10% of the sample that was incubated with the GST-RhoAG17A probe, or GST only for the control. Coomassie blue stained gel (lower panel) showing lysates, GST-Rho-G17A (blue arrow), and GST (red arrow) in the pulldown. The box indicates the position of GEF-H1. (**B**) Active GEF-H1 levels increased during antigen stimulation. An immunoblot of active GEF-H1 from a GST-RhoA-G17A pulldown (upper panel) and 10% load control (lower panel) of RBL-2H3 cell lysates prepared after antigen stimulation. Active GEF-H1 levels were quantified by band densitometry of immunoblots and statistically analyzed by one-way ANOVA (*p* = 0.0024; *n* = three independent blots). Tukey’s tests were used to identify significant differences in pairwise comparisons between stimulation times (** *p* = 0.0018 comparing 0 vs. 20 min stimulation). (**C**) GEF-H1 activation was not affected by microtubule-targeted drugs. An immunoblot of active GEF-H1 (upper panel) and total GEF-H1 from a 10% load control (lower panel) of RBL-2H3 cell lysates antigen-stimulated for 0 min (Unstim) or 20 min (Stim), or stimulated 20 min after pretreatment with 10 µM of taxol (Taxol) or 10 µM of nocodazole (Noco). Active GEF-H1 levels were quantified by band densitometry of immunoblots and statistically analyzed by one-way ANOVA (*p* = 0.0046; *n* = three independent blots). Tukey’s tests were used to identify significant differences in pairwise comparisons between stimulation conditions (** *p* = 0.0018 comparing 0 vs. 20 min stimulation; ns, not significant comparing GEF-H1 activation levels in stimulated cells vs. pretreatment with taxol or nocodazole). (**D**) Tubulin was not detected in GEF-H1 immunoprecipitates. RBL-2H3 cells were stimulated for 0–20 min, lysed, and GEF-H1 was immunoprecipitated using a GEF-H1-specific antibody. Immunoblots show no β-tubulin in GEF-H1 IP fractions.

**Figure 9 cells-12-00537-f009:**
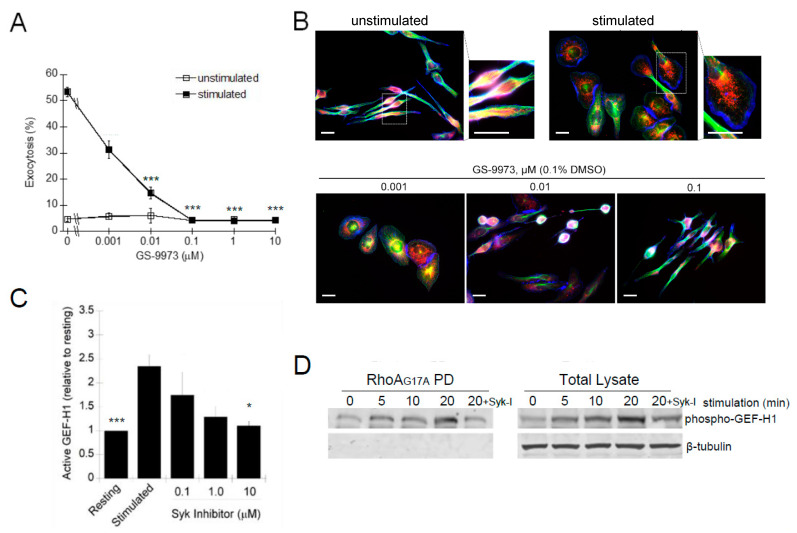
GEF-H1 activation is regulated by Syk-mediated phosphorylation. RBL-2H3 cells were preincubated for 20 min with 487 varying concentration of the Syk inhibitor, GS-9973, and its effect on antigen-stimulated exocytosis was examined by degranulation 488 assay and microscopy. (**A**) GS-9973 significantly reduced RBL-2H3 cell granule exocytosis (*** *p* < 0.001 compared to DMSO control 489 by one-way ANOVA and post hoc Tukey’s test; *n* = 3 independent assays). (**B**) Images of cells preincubated with GS-9973 show 490 reduced cell spreading and granules remained in perinuclear regions. Cells were fixed and stained for granules with anti-CD63 491 antibody (red), microtubules with β-tubulin antibody (green) and F-actin with Alexa Fluor 405-phalloidin (blue). Scale bar = 20 μm. 492 (**C**,**D**) Active GEF-H1 levels were determined by GST-RhoA-G17A pulldown assays after antigen-stimulated for 20 min. (**C**) 493 GS-9973 reduced GEF-H1 activation in antigen-stimulated RBL-2H3 cells. Statistical analysis was performed by one-way ANOVA 494 (* *p* < 0.001; *n* = 3 independent blots). Tukey’s tests were used to identify significant differences in pairwise comparisons (*** *p* < 0.001 495 resting vs stimulated; * *p* = 0.034 stimulated vs 10 μM Syk inhibitor). (**D**) Immunoblot of pulldown fractions and total lysates with 496 phospho-GEF-H1 (Ser886) antibody shows active GEF-H1 is phosphorylated with levels increasing up to 20 min of antigen stimula-497 tion; GS-9973 (Syk-I) reduced the levels of phospho-GEF-H1 in total lysates and pulldown fractions.

## Data Availability

The data presented in this study are openly available in FigShare at DOI: 10.6084/m9.figshare.21453942.

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
