# Peer review of "GEF-H1 Transduces FcεRI Signaling in Mast Cells to Activate RhoA and Focal Adhesion Formation during Exocytosis"

_cells, 2023, doi:10.3390/cells12040537_

Round 1
Reviewer 1 Report
GEF-H1 silencing impairs cell spreading and degranulation. The authors investigated the role of the RhoGEF, GEF-H1/ARHGEF2, in IgE-dependent degranulation in RBL-2H3. The mechanism involved RhoA and Syk phosphorylation of GEF-H1. The authors conclude that GEF-H1-RhoA signaling is critical for mast cell exocytosis. The work is well performed and exposed.
The weak points of the paper are that all the experiments are performed in a single cellular model (RBL.2H3), and only one sequence for silencing is used. It would have been nice showing the relevance of this signaling in another cellular model. Regarding silencing, they reconstitute with RhoGEF and RhoA and revert the RhoGEF silencing effects showing specificity. RhoGEF and RhoA overexpression restores spreading, but authors should show what happens in degranulation, which is the main outcome they are searching for.
Pak2 regulates GEF-H1 through phosphorylation to negatively control mast cell degranulation (Kosoff et al. JBC 2013). It should be mentioned and discussed.
Minor points
Scale bars in the images are missing in some figures, e.g., figs 2, 3...Please check.
Author Response
Are all the cited references relevant to the research? <can be improved>
We will carefully review our references.
GEF-H1 silencing impairs cell spreading and degranulation. The authors investigated the role of the RhoGEF, GEF-H1/ARHGEF2, in IgE-dependent degranulation in RBL-2H3. The mechanism involved RhoA and Syk phosphorylation of GEF-H1. The authors conclude that GEF-H1-RhoA signaling is critical for mast cell exocytosis. The work is well performed and exposed.
The weak points of the paper are that all the experiments are performed in a single cellular model (RBL.2H3), and only one sequence for silencing is used. It would have been nice showing the relevance of this signaling in another cellular model.
For this study we did use only one mast cell model, RBL-2H3 cells, as this is the only immortalized mast cell that responds to FceRI ligands. Primary mast cells from rat bone marrow or mouse bone marrow could have been used but we would need to obtain a GEF-H1 KO mouse to do these experiments (a conditional KO would be even better). Unfortunately, this is beyond the resources available to us. In the discussion we will add a section about the limitations of our study and the need to validate findings in another cellular model or animal system [line 265]. We will also discuss previous findings that have shown a role of GEF-H1 in exocytosis in other cellular models, however this would be less relevant to Fc signaling [lines 520 and 538]. Note that GEF-H1 has been shown to control exocytosis [Pathak 2012, PMID: 22898781]. However, this study was done in HeLa cells and not examining physiologically relevant processes. A more recent study shows GEF-H1 controls exocytosis at the immune synapse which is a highly relevant physiological process [Saez 2019, PMID: 31197029].
Regarding silencing, they reconstitute with RhoGEF and RhoA and revert the RhoGEF silencing effects showing specificity. RhoGEF and RhoA overexpression restores spreading, but authors should show what happens in degranulation, which is the main outcome they are searching for.
We cannot do the degranulation assay on transfected cells. This is an enzyme assay on whole cell secretion and since only ~15% of RBL-2H3 cells take up the plasmids most cells would not show the effect of the plasmid. So since transfection efficiency for RBL-2H3 cells is low, little change would be observed.
Pak2 regulates GEF-H1 through phosphorylation to negatively control mast cell degranulation (Kosoff et al. JBC 2013). It should be mentioned and discussed.
Yes, GEF-H1 can be both activated and deactivated via differential phosphorylation. A more extensive analysis of the control of GEF-H1 by phosphorylation will be added to the discussion, in particular negative regulation by Pak2 phosphorylation [line 551].
Minor points
Scale bars in the images are missing in some figures, e.g., figs 2, 3...Please check.
Scale bars have been added to all images.
Reviewer 2 Report
The authors have investigated the roles of the Rho family in activated mast cells. Here they focused on the functional roles of a RhoGEF, GEF-H1, using a rat mast cell line, RBL-2H3, and its derivative with lower expression levels of GEF-H1. The study was well-designed and carefully performed. They demonstrated that IgE-mediated antigen stimulation should activate GEF-H1, which induces enhanced focal adhesion through RhoA activation. I believe this study should be significant for researchers in this field. I would like to raise several concerns that should be addressed before publication.
The levels of degranulation did not reach the plateau even 30 minutes after the antigen stimulation here (Fig. 1C). However, accumulating evidence indicated that degranulation of mast cells including RBL-2H3 is usually completed within 30 minutes after the initial stimulus. The authors performed the morphological analyses 20 to 30 minutes after the antigen stimulation. Degranulation might terminate at these time points. This time course may suggest that GEF-H1-mediated morphological changes occur at the recovery phase of degranulated mast cells and are not involved in activated exocytosis. It still remains unknown how GEF-H1 could enhance degranulation.
The authors used both an anti-CD63 antibody and LysoTracker to identify the granules of mast cells. The signals of LysoTracker disappear when exocytosis occurs, whereas CD63 remains in the plasma membrane of degranulated mast cells. The signals of CD63 accumulate in the plasma membrane upon mast cell activation. How did the authors utilize these two different granule markers here?
Some fluorescence images, such as Figs. 3A, 3B, 6A, 7B, and 9B, need good resolutions because it might be challenging to confirm the author's statement based on these images.
The authors should describe in detail how they established GEF-H1-depleted RBL-2H3 cells. Is this a permanent clone, in which the expression of GEF-H1 is stably suppressed by RNAi?
Author Response
Are the methods adequately described? <can be improved>
More fully described methods will be added to the revised manuscript.
Are the conclusions supported by the results? <can be improved>
Conclusions drawn from results will be more carefully described. In particular, we have changed the title and removed “for exocytosis” and suggest that GEF-H1 function is needed “during exocytosis” and facilitates the process but may not be directly part of the exocytosis mechanism.
The authors have investigated the roles of the Rho family in activated mast cells. Here they focused on the functional roles of a RhoGEF, GEF-H1, using a rat mast cell line, RBL-2H3, and its derivative with lower expression levels of GEF-H1. The study was well-designed and carefully performed. They demonstrated that IgE-mediated antigen stimulation should activate GEF-H1, which induces enhanced focal adhesion through RhoA activation. I believe this study should be significant for researchers in this field. I would like to raise several concerns that should be addressed before publication.
The levels of degranulation did not reach the plateau even 30 minutes after the antigen stimulation here (Fig. 1C). However, accumulating evidence indicated that degranulation of mast cells including RBL-2H3 is usually completed within 30 minutes after the initial stimulus. The authors performed the morphological analyses 20 to 30 minutes after the antigen stimulation. Degranulation might terminate at these time points. This time course may suggest that GEF-H1-mediated morphological changes occur at the recovery phase of degranulated mast cells and are not involved in activated exocytosis. It still remains unknown how GEF-H1 could enhance degranulation.
In the experimental conditions we use, RBL-2H3 exocytosis levels increase ~50% between 15 min and 30 min and rise ~10% more between 30 min and 45 min. This is why we often do our analyses between 20 and 30 min; this is approximately when we find exocytosis rates are at their peak.
One reason for this may be that cells do not respond synchronously. This asynchronous response can be seen in the following linked live cell video where the cells respond to antigen stimulation over the course of the 20 min video (30x): (RBL-2H3 stimulation). We found RBL-2H3 cells have to form “exocytosis zone” first and then exocytosis rates are maximized [Sheshachalam 2017, PMID: 28411215].
However, we do have results that suggest RhoA functions after Rac1 and Rac1 is primarily needed for the activation phase of exocytosis, while RhoA may function to maintain the active state. So, we do agree with this point and we will consider that RhoA activation could be initiating a recovery phase and residual degranulation. This possible interpretation of the data will be included in the discussion [line 525]. We will tone down our conclusion that GEF-H1 IS needed for degranulation. For example, in the title “during exocytosis” and in the abstract instead of concluding GEF-H1 IS required for degranulation we say “We concluded that GEF-H1 acts to transmit signals from surface receptor stimulation to RhoA activation and FA formation during mast cell exocytosis.”
The authors used both an anti-CD63 antibody and LysoTracker to identify the granules of mast cells. The signals of LysoTracker disappear when exocytosis occurs, whereas CD63 remains in the plasma membrane of degranulated mast cells. The signals of CD63 accumulate in the plasma membrane upon mast cell activation. How did the authors utilize these two different granule markers here?
Lysotracker was used in live cells and the images are frames taken from the videos. When cells are fixed we stain granules with antibodies to CD63. Our previous work confirms that lysotracker co-localizes with CD63 labelling [Ibanga, PMID: 35316306]. These labels have been used to mark granules in our work in several publications [Ibanga 2022, PMID: 35316306; Sheshachalam 2017, PMID: 28411215] as well as by other labs [Masuda 2000 PMID: 10722846, Amano 2001 PMID: 11328596, Köberle 2012 PMID: 22783251;,Ikeya 2014 PMID: 25044118].
Some fluorescence images, such as Figs. 3A, 3B, 6A, 7B, and 9B, need good resolutions because it might be challenging to confirm the author's statement based on these images.
The insertion of images into the document (a word file) resulted in significant loss of resolution. We have inserted improved images in the resubmission with much improved resolution.
The authors should describe in detail how they established GEF-H1-depleted RBL-2H3 cells. Is this a permanent clone, in which the expression of GEF-H1 is stably suppressed by RNAi?
The GEF-H1 knock-down strain was a lentiviral (retroviral) transduction that stably integrated the shRNA into the genome of the cell, allowing for persistent expression. We will improve our methods section and provide more details on how reagents were developed and how experiments were performed.
Round 2
Reviewer 1 Report
About the question
RhoGEF and RhoA overexpression restores spreading, but authors should show what happens in degranulation, which is the main outcome they are searching for.
The answer of the authors
"We cannot do the degranulation assay on transfected cells. This is an enzyme assay on whole cell secretion and since only ~15% of RBL-2H3 cells take up the plasmids most cells would not show the effect of the plasmid. So since transfection efficiency for RBL-2H3 cells is low, little change would be observed".
Still, there are some concerns. Problems of transfection do not affect to see significant differences in cell spreading, just in degranulation assays? It should be the same problem. If the levels of transfection are so low, both readouts should be affected. How do you monitor transfected cells versus non-transfected ones in the cell-spreading assays? All of these needs justification. On the other hand, RBL-2H3 are easy models to transfect and are widely reported in the literature. What problems exactly do the authors face?
If the authors do not show consistent data about degranulation, they should conveniently modify the title and conclusions.
